# Clinical Phenotypes and Mortality Biomarkers: A Study Focused on COVID-19 Patients with Neurological Diseases in Intensive Care Units

**DOI:** 10.3390/bs12070234

**Published:** 2022-07-15

**Authors:** Lilia María Morales Chacón, Lídice Galán García, Tania Margarita Cruz Hernández, Nancy Pavón Fuentes, Carlos Maragoto Rizo, Ileana Morales Suarez, Odalys Morales Chacón, Elianne Abad Molina, Luisa Rocha Arrieta

**Affiliations:** 1International Center for Neurological Restoration, Havana 11300, Cuba; nancy.pavon@infomed.sld.cu (N.P.F.); subdirectorclinico@neuro.cu (C.M.R.); elianneamolina@gmail.com (E.A.M.); 2Cuban Neurosciences Center, Havana 11300, Cuba; lidicegalan2000@gmail.com; 3Cuban Ministry of Health, La Habana 4JR9+68J, Cuba; vmas@infomed.sld.cu (T.M.C.H.); imorales@infomed.sld.cu (I.M.S.); 4Languages Center, Technological University of Havana Jose Antonio Echeverria, La Habana 3H3M+XJ6, Cuba; odalysmoraleschacon@gmail.com; 5Center for Research and Advanced Studies México, Ciudad de México 14330, Mexico; lrocha@cinvestav.mx

**Keywords:** COVID-19, clinical phenotypes, intensive care units, comorbidities, neurological diseases, stability-based biomarkers

## Abstract

**Purpose:** To identify clinical phenotypes and biomarkers for best mortality prediction considering age, symptoms and comorbidities in COVID-19 patients with chronic neurological diseases in intensive care units (ICUs). **Subjects and Methods:** Data included 1252 COVID-19 patients admitted to ICUs in Cuba between January and August 2021. A k-means algorithm based on unsupervised learning was used to identify clinical patterns related to symptoms, comorbidities and age. The Stable Sparse Classifiers procedure (SSC) was employed for predicting mortality. The classification performance was assessed using the area under the receiver operating curve (AUC). **Results:** Six phenotypes using a modified v-fold cross validation for the k-means algorithm were identified: phenotype class 1, mean age 72.3 years (ys)—hypertension and coronary artery disease, alongside typical COVID-19 symptoms; class 2, mean age 63 ys—asthma, cough and fever; class 3, mean age 74.5 ys—hypertension, diabetes and cough; class 4, mean age 67.8 ys—hypertension and no symptoms; class 5, mean age 53 ys—cough and no comorbidities; class 6, mean age 60 ys—without symptoms or comorbidities. The chronic neurological disease (CND) percentage was distributed in the six phenotypes, predominantly in phenotypes of classes 3 (24.72%) and 4 (35,39%); χ² (5) 11.0129 *p* = 0.051134. The cerebrovascular disease was concentrated in classes 3 and 4; χ² (5) = 36.63, *p* = 0.000001. The mortality rate totaled 325 (25.79%), of which 56 (17.23%) had chronic neurological diseases. The highest in-hospital mortality rates were found in phenotypes 1 (37.22%) and 3 (33.98%). The SSC revealed that a neurological symptom (ageusia), together with two neurological diseases (cerebrovascular disease and Parkinson’s disease), and in addition to ICU days, age and specific symptoms (fever, cough, dyspnea and chilliness) as well as particular comorbidities (hypertension, diabetes and asthma) indicated the best prediction performance (AUC = 0.67). **Conclusions**: The identification of clinical phenotypes and mortality biomarkers using practical variables and robust statistical methodologies make several noteworthy contributions to basic and experimental investigations for distinguishing the COVID-19 clinical spectrum and predicting mortality.

## 1. Introduction

The clinical spectrum and severity of COVID-19 are broad and complex. Additionally, the clinical presentation and severity of COVID-19 patients can vary widely and extend outside the respiratory system. Some individuals are asymptomatic, whereas others may develop severe pneumonia with acute respiratory failure [1,2,3].

Intensive care units (ICUs) have an important role in managing the sickest of these patients [4,5]. Nevertheless, mortality is prominently high in this group. A systematic review and meta-analysis of 52 observational studies [6] published by April 2021, which included 43,128 COVID-19 patients in ICUs, revealed that in most geographical regions, the mortality rate was 30–40% [4].

Several chronic conditions have proven to be associated with a more severe disease and death, such as heart diseases, hypertension, diabetes mellitus, chronic pulmonary disease, obesity and cancer [7,8,9]. Other conditions, including asthma, pregnancy and slight immunosuppressant conditions are affected by mixed or more limited evidence. Illnesses such as chronic renal diseases and cytokine activation syndrome may be exacerbated due to COVID-19, and extra care is required for such patients [10].

Although most COVID-19 patients develop primarily respiratory symptoms, the occurrence of neurological symptoms in 30–50% of patients has been associated with disease severity and mortality, suggesting a potential neurotropism of SARS-CoV-2 as a possible mechanism of neurological damage [11,12]. Moreover, some studies have indicated that neurological disease may be a risk factor for COVID-19, in part because of the higher rate of comorbidities associated with these patients. However, the frequency, type and implications of chronic neurological diseases (CNDs) in patients with COVID-19 have been little explored and are still unknown.

Even though a large number of investigations have been conducted around the world in a very short period, several problems remain unsolved, including the clinical and biological heterogeneity of COVID-19. This has led to incomplete categorization of disease phenotypes and hampered stratification of key patients in pandemic control. Cluster analysis has been used to investigate the heterogeneity of some diseases in order to identify clinical phenotypes with similar trait mixtures. A previous research approach in asthma, cancer and acute respiratory distress syndrome was able to identify disease phenotypes with significant therapeutic implications [13,14,15].

In this study, we used a modified v-fold cross validation for using the k-means algorithm and the Stable Sparse Classifiers (SSC) described by Bosch (2018) to solve the classification problem based on sparse regressions combined with techniques for ensuring stability, which is especially useful for high-dimensional datasets and small sample numbers [16]. The sensitivity and specificity of the classifiers were assessed by a stable receiver operating characteristic (ROC) procedure, which employs a non-parametric algorithm for estimating the area under the ROC curve. 

This article aims at identifying clinical phenotypes and biomarkers for best mortality prediction considering age, symptoms and comorbidities in COVID-19 patients with chronic neurological diseases in ICUs.

## 2. Subjects and Methods

Data were gathered from 1252 ICU patients with COVID-19 in Cuban hospitals. The diagnosis was confirmed by positive high-throughput sequencing or a real-time reverse-transcription polymerase chain reaction (RT-PCR) assay for nasal and pharyngeal swab specimens.

Symptoms and comorbidities of all patients were collected upon admission. Then, patients were assessed with clinical, radiological and laboratory examinations during the first 24 h of ICU admission. Lastly, collection of clinical data from each patient, including treatments and outcomes, were verified by the leading researcher LMCh.

Demographic variables included age, sex and race. Clinical symptoms, on the other hand, encompassed fever, cough, nasal congestion, headache, fatigue, dyspnea, rhinorrhea, nausea/vomiting, diarrhea, arthralgia, myalgia, ageusia, anosmia, chilliness, chest pain and unconsciousness.

Regarding comorbidities, we analyzed the presence of hypertension, described as systolic blood pressure (SBP)/diastolic blood pressure (DBP) ≥140/90 mmHg and/or the use of antihypertensive agents, diabetes, defined as fasting blood glucose ≥126 mg/dL on two occasions and/or the use of antidiabetic agents, smoking habit (current or in the preceding 6 months), coronary artery disease, chronic obstructive pulmonary disease (COPD), asthma, cancer, obesity, psychiatric diseases, chronic renal diseases, alcoholism, pregnancy, immunocompromised state (congenital or acquired) and chronic neurological disorders (CNDs). 

According to the World Health Organization, disabling CNDs can be defined as those neurological disorders that (a) cause persistent disability, (b) limit the individual’s functioning and (c) interfere with the person’s ability to perform in activities (20). In addition to these characteristics, conditions affecting mental and physical functions were also included. Hence, CNDs include dementia, epilepsy, movement disorders, previous stroke with long-term sequelae, neuromuscular disorders, spinal disorders, symptomatic central nervous system cancer, chronic encephalopathies and neuroinflammatory disorders.

Imaging results involved a chest radiography (CXR) abnormality assessment. Laboratory examinations, on the other hand, included white blood, lymphocytes and neutrophil count, platelet number, mean platelet volume, and neutrophil to lymphocyte ratio. Other variables embraced included ventilation type and illness severity at ICU admission. 

Patients were treated in accordance with the Cuban protocol version 1.6, available over the pandemic period https://covid19cubadata.github.io/protocolos/protocolo-version-6.pdf (accessed on February 2022). The primary outcome of the study was in-hospital mortality. 

The study was approved by the Innovative Commission of the Cuban Health Ministry, as it followed the Ethical Principles for Medical Research Involving Human Subjects established by the Declaration of Helsinki of 1975 for human research. Written informed consent was waived owing to the use of identified retrospective data. 

### 2.1. Statistical Analysis

Qualitative and ordinal variables are indicated in terms of frequency and percentage. Continuous variables are shown as medians, minimum-maximum value or mean and standard deviation (SD).

Statistical analysis involved the chi-square test or Fisher’s exact test for contrasting categorical variables, adjusting the p-value by the Bonferroni method for multiple comparison correction. Due to the exploratory nature of the study, the sample size was not calculated. Furthermore, the level of significance threshold was set as 0.05 after adequate adjustment.

#### 2.1.1. k-Means Algorithm

The k-means algorithm based on unsupervised learning was used [17]. The optimal number of six clusters (k) was estimated using a modified v-fold cross validation for using the k-means algorithm.

The steps of the algorithm include:Determining the number of clusters k;Setting centroids by first shuffling the dataset, and then randomly selecting data points to replace the centroids;Calculating the distance between data points and all centroids;Allocating each data point to the closest cluster (centroid);Updating the position of the centroid according to the assigned data;Retaining iteration until there is no change to the centroids.

At this point, the values are assigned into k clusters without any hierarchical structure by optimizing the minimum distance between points in each of the available clusters and applying the Euclidean distance between data points and centroids as a distance criterion.

All analyses were completed using statistics software (version 12). Modules utilized for the analyses encompassed the Generalized EM and k-Means Cluster Analysis, included in the data mining.

#### 2.1.2. Stable Sparse Biomarkers Detection: The Procedure to Select a Stable and Sparse Classifier 

Biomarkers selection: Initially, data were perturbed (70% of subjects and 70% of variables were randomly selected using cross-validation), followed by a screening step for eliminating variables with a minimal contribution. Later, a smaller set of biomarkers was selected using glmnet with cross-validation. These steps were repeated (n = 500) to identify the variables present at least 50% of the time. Then, parameters of the models were calculated intrinsically by the procedure.

Model validation (using the variables previously selected) based on ROC values and stability assessments: A random subsample with 70% of the subjects was used to classify using glmnet, and the rest of the sample was used to calculate the ROC values. This procedure was executed in several iterations (n = 500) to estimate the distribution of ROC values. The AUC at the 50th percentile of the distribution was utilized to measure classification accuracy, and the model with the highest accuracy was then selected as the best model.

Model formulation: glmnet.

The elastic-net model used to select the biomarkers was formulated as a weighted multivariate linear regression model described by the equation:minβ0∈ℝ, β∈ℝp[12N∑i=1N(yi−φ0−xiTφ)2+λ Pγ (φ)]
where ***N*** is the number of subjects, the number of observations of subject ***i*** and the label group of subject *i*, which are the model parameters; the regularization parameter; *p* is the number of variables in the model; and Pγ (φ)=(1−γ)12 ‖φ‖l22+γ ‖φ‖l1 is the penalty equation known as the elastic-net norm. The use of the L2 norm induces a regression (known as ridge regression) that behaves well for high-dimensional data but tends to spread out coefficient weights among highly correlated variables, and the L1 norm produces the “lasso regression”, indifferent to highly correlated predictors, which tries to estimate only a few nonzero coefficients (i.e., performs variable selection), thus inducing sparsity in the vector of coefficients. They are the parameters of the relative contributions of the ridge and the lasso to the elastic net. 

The characteristics of the elastic net allow SSC to deal with highly correlated features. In addition, the glmnet classification algorithm can solve the problems related to the high number of features and the low number of subjects. These are two useful advantages for this study. 

## 3. Results

The mean age of ICU patients was 66.13 ± 16.33 years (age range: 19–104 years old); 717 (57.26%) were male. The most frequent comorbidities were hypertension (895, 71.48%), diabetes (373, 29.79%) and cardiovascular disease (309, 24.68%). No atrial fibrillation was identified in these patients. On the other hand, dry cough (578, 46.17%), fever (430, 34.34) and dyspnea (350, 27.95) were acknowledged to be the most common presenting symptoms. According to the Cuban Emergency Society guideline, 225 (18%) were considered to be critically ill at ICU admission.

From the total sample, at least one CND was present in 197 patients (15.74%). The average age of patients with chronic neurological disease was 73.84 ± 15.15. One hundred and twenty-two (56.4%) were male. Although patients with CNDs were older, the rates of hypertension, coronary artery disease and diabetes were similar to those of patients with no CNDs. In terms of symptoms, there were no differences between the two groups.

The most common chronic neurological disease was dementia, which was present in 72 cases of the patient group (36.55%), followed by cerebrovascular disease in 68 cases (34.52%), epilepsy in 33 (16.75%) and PD in 13 (6.6%). 

Additionally, most patients (88.8%) had abnormal chest radiography (CXR) findings. With regard to blood biomarkers, a neutrophil to lymphocyte ratio >4 was observed in 38.23% of the patients, and mean platelet volume >9 in 36.71%.

Concerning treatment, 52.28 % of the patients received steroids, antibiotics (55.92%), anticoagulants (53.97%) and Jusvinza (Péptido CIGB (77.6%). AcMc (itolizumab) and nimotuzumab were also used in 150 of the cases. 

### 3.1. Characteristics of k-Means Algorithm-Defined Phenotypes for COVID-19 Patients in ICUs Considering Fifteen Symptoms, Fourteen Comorbidities and Age

The optimal number of six clusters (k) was estimated using a modified v-fold cross-validation for the use of the k-means algorithm. One demographic variable (age), fifteen symptom variables and fourteen comorbidity variables were used to define clinically the six phenotypes below: ✓ Phenotype class 1: mean age 72.3 years—hypertension, coronary artery disease, cough and fever; n = 222 (17.74% of the sample);✓ Phenotype class 2: mean age 63 years—asthma, cough and fever; n = 67 (5.3% of the sample);✓ Phenotype class 3: mean age 74.5 years—hypertension, diabetes and cough; n = 255 (20.3% of the sample);✓ Phenotype class 4: mean age 67.8 years—hypertension and no symptoms; n = 394 (31.4% of the sample);✓ Phenotype class 5: mean age 53 years—cough and no comorbidities; n = 123 (9.8% of the sample);✓ Phenotype class 6: mean age 60 years—without symptoms and comorbidities; n = 191 (15.2% of the sample);

Comorbidities were more common in the mean age of 70 years than in younger adult classes (phenotypes 5 and 6). Phenotype classes 1 and 3 combined two comorbidities and comprised individuals with median ages 72 and 74 years, respectively. Conversely, phenotype classes 5 and 6 embraced the mean age of 50 year old patients.

### 3.2. Defined Phenotypes for COVID-19 Patients with Pre-Existing Neurological Diseases in ICUs 

The percentage of CNDs was distributed across the six phenotypes, predominantly in phenotype classes 3 (24.72%) and 4 (35.39%). For the rest of the phenotypes, the distribution was as follows: class 1 (12.9%), class 2 (6.18%), class 5 (5.62%) and class 6 (15.17%). See Figure 1.

There was a significant association with the occurrence of CNDs and the k-means algorithm-defined phenotypes; χ² (5) 11.0129, *p* = 0.051134. Moreover, the cerebrovascular disease was concentrated in classes 3 and 4. Hence, an important relationship between the presence of cerebrovascular diseases and the k-means algorithm-defined phenotypes was detected; χ² (5) = 36.63, *p* = 0.000001.

### 3.3. Defined Phenotypes and Mortality Rate in COVID-19 Patients with and without Neurological Diseases in ICUs

The mortality rate of the sample totaled 325 (25.79%). The chi-square χ² analysis indicated a relationship between the cluster analysis-defined phenotypes, illness severity and inpatient mortality, where χ² (5) = 12.601, *p* = 0.027 and χ² (5) = 51.211, *p* = 0.000, respectively. The inpatient mortality rates for the six phenotypes were 37.22%, 14.93%, 33.98%, 22.28%, 18.55% and 13.02%. The highest in-hospital mortality rates were found in phenotypes 1 and 3. In contrast, phenotypes 6 and 2 displayed the lowest fatality rates. Furthermore, the ratio between deaths and survival in the cohort was 0.34: cluster 1 (0.59), cluster 2 (0.17), cluster 3 (0.51), cluster 4 (0.28), cluster 5 (0.22) and cluster 6 (0.14). As can be observed, only phenotypes 1 and 3 exceeded the overall rate of the cohort.

With reference to chronic neurological diseases, 56 (17.23%) deaths were reported. About 56% involved cerebrovascular disease and 39% involved dementia, followed by epilepsy at 7.32% and PD at 4.88%. Comparable to the mortality rate associated with the six phenotypes defined for COVID-19, most of the patients with CNDs who died were grouped in clusters of class 3 (34.15%), class 1 (29.27%) and class 4 (24.3%); and to a lesser extent in classes 5 (4.8%) and 6 (7.32%). No deaths were reported in patients with CNDs in cluster 2. 

### 3.4. Stable Sparse Classifiers Procedure (SSC) Based on Mortality Prediction in COVID-19 Patients in ICUs 

Neurological variables including ageusia, together with cerebrovascular disease and Parkinson’s disease and in addition to age, ICU days and specific symptoms such as fever, cough, dyspnea and chilliness as well as comorbidities comprising hypertension, diabetes and asthma were selected as mortality biomarkers in COVID-19 patients in ICUs. Figure 1 shows the results of the classification procedure. The constructed model includes the variables of the 1252 patients divided into three groups. The procedure identifies 20 variables which can be considered as biomarkers. Figure 1 shows the weight associated with each variable, accordingly. The classification performance is reflected in Figure 2A through the area under the curve (AUC = 0.67). This reflects the heterogeneity behavior in the groups. However, Group 1 (deaths) can be differentiated from Groups 2 and 3 (no deaths), i.e., Group 2 (critically ill) and Group 3 (serious condition), in relation to the scores obtained in the procedure (Figure 2B).

## 4. Discussion

The current study used a large cohort of COVID-19 patients in ICUs, focused on an objectively classifying diseases into clinical phenotypes. We defined six clinical phenotypes based on age, ICU days, symptoms and comorbidities using a modified v-fold cross-validation for the use of the k-means algorithm. Furthermore, the Stable Sparse Classifiers procedure allowed identification of twenty biomarkers to predict mortality.

In this study, phenotypes 1 and 3, including elderly people with respiratory symptoms alongside two different comorbidity patterns, hypertension either with diabetes or with artery coronary disease, emerged as important clinical phenotypes. On the other hand, phenotype class 4 was characterized by hypertension, no symptoms and a mean age of 67.8 years. We showed that advanced age and hypertension associated with diabetes or with coronary artery disease was strongly related to a high rate of in-hospital mortality in ICU patients. 

Recently, some approaches to phenotype clusters of COVD-19 have been described [18,19]. One recent study developed a predictive equation based on symptoms, comorbidities and demographic data. These variables provided fair ability to discriminate severe vs. non-severe outcomes [20]. 

The current study demonstrated that the phenotypic cluster along with the Stable Sparse Classifiers approach is a better predictor of mortality in ICUs than risk assessment merely based on age, symptoms and comorbidities. This provides an incremental benefit to our prediction model over existing knowledge that is also concordant with the country-wide pattern of COVID-19-associated comorbidities in hospitalized patients and deaths. Heart diseases, including hypertension in conjunction with cardiovascular diseases, are the most frequent associations with SARS-CoV2 infection in most countries [21]. 

Similar reports of high prevalence of cardiovascular diseases and hypertension among hospitalized patients were demonstrated in case reports from China and the United States [22,23,24]. In other studies, approximately 25% of COVID-19 cases had at least one related comorbidity [25,26,27,28]. 

The findings from this study also suggest other potentially relevant clinical implications. For example, phenotype class 2 (mean age 63, asthma, cough and fever), phenotype class 6 (mean age 60, no symptoms and no comorbidities) and phenotype class 5 (mean age 53, cough and no comorbidities) showed the highest survival probability. These results may be considered among the highest rates of asymptomatic COVID-19 patients reported in the literature to date [29]. Moreover, a young age was found to be the only factor associated with an asymptomatic course [30]. 

As to respiratory symptoms in the phenotypes, the results of this study support previous research in which cough, dyspnea and/or fever were considered to be the most common symptoms [18,19]. 

Overall, the death rate of COVID-19 patients in ICUs in the current study was 25.79%, which is lower than in other countries [6]. Yet, this figure is relevant when compared to the usually described mortality rates of community-acquired pneumonia of about 16.6–18% [31,32]. It is worth mentioning that the Cuban health care system provides the elderly population with a broad range of health services. It also stresses prevention and control of chronic diseases from primary healthcare, which is crucial to mitigate the impact of severe COVID-19 and fatality on this age group [33].

In this study, the prevalence of CNDs in COVID-19 patients in ICUs was approximately 15.7%. Furthermore, patients with CNDs were older, and approximately 30% had diabetes or coronary artery disease. As a whole, symptoms in this group were similar to those of subjects with no CNDs. These results are consistent with a cohort study in which patients with CNDs were older, more disabled, had more vascular risk factors and comorbidities, and had the same need for ventilatory support [34]. A recent PRISMA guideline-based systematic review revealed that headache, musculoskeletal injuries, psychiatric disorders, altered consciousness and gustatory/olfactory dysfunction were the most common neurological symptoms in COVID-19 patients. Besides, altered consciousness and acute cerebrovascular events were significantly higher among patients with severe infection [35]. 

Additionally, the mortality rate in CND was around 17.23%, which is reasonably low and most likely multifactorial. Therefore, the potential neurotropism of SARS-CoV-2 with a detrimental effect on chronic neurological diseases, advanced age and the higher burden of comorbidity observed in these patients should be considered [36,37].

This study demonstrated a relationship between chronic neurological disease, clinical phenotypes and in-hospital mortality in patients with COVID-19. CNDs were distributed among the six phenotypes (according to age, symptoms and comorbidities), predominantly in phenotypes 3, 2 and 4. This finding offers insights into the clinical spectrum of COVID-19 patients in ICUs, comprising distinct phenotypes based on patients’ clinical information available during hospital admission. A number of previous studies identified isolated predictors of severe disease progression, and developed clinical risk scores, but such information has limited impact on clinical practice [38,39,40].

This paper presents a prediction method to achieve stable classifiers for clinical and demographic variables using a minimal set of clinical features (biomarkers) to predict the mortality in patients with COVID-19 in ICUs.

The SSC revealed that a neurological symptom (ageusia) accompanied by two neurological diseases (cerebrovascular disease and PD), in addition to ICU days, age and specific symptoms (fever, cough, dyspnea and chilliness) as well as particular comorbidities (hypertension, diabetes and asthma) indicated the best prediction performance.

There is some evidence based on the SARS and MERS epidemics that these comorbidities predispose risk factors for severe infections, leading to critical care and fatality. The same trend has been observed with SARS-CoV-2 and COVID-19 disease [38,41,42,43]. 

Our results corroborate the notion that COVID-19 is associated with various comorbidities often related to endothelial dysfunction, such as hypertension, diabetes and neurological diseases [21,44], indicating that the endothelium may be targeted by SARS-CoV-2. It was demonstrated that diabetes is a significant risk factor for hospitalization and adverse outcomes in these patients [45,46] and is also associated with an increased risk of thromboembolism among COVID-19 patients [47].

This finding underlines the relevance of the association between age, hypertension, diabetes and other CND comorbidities in in-hospital mortality. Jakhmola et al. reported that losses due to renal diseases and neurological conditions are significantly higher than the total of hospitalized patients affected by specific comorbidity [21]. In addition, a systematic review and meta-analysis also concluded that advanced age and pre-existing comorbidities were related to mortality [48]. 

A fundamental area of concern for neurologists is how COVID-19 affects patients with underlying neurological diseases. Interestingly, we found a higher mortality rate in patients with cerebrovascular disease, followed by dementia and epilepsy. Moreover, one report demonstrated that the presence of CND is an independent predictor of mortality in patients hospitalized for COVID-19 [49]. We assumed that, such as in the case of SARS and MERS, the Angiotensin-converting enzyme-2 (ACE2) receptors may play a crucial role in determining the severity of the disease, as well as the upregulation of ACE2 [50,51,52].

Concerning cerebrovascular disease, the findings in this study mirror those reported in two meta-analyses by Wang et al. [53] and Aggarwal et al. [54], which showed that underlying cerebrovascular disease is related to higher disease severity in patients with COVID-19. These results are also consistent with a recent study by Romagnolo et al., who stated that cerebrovascular disease and cognitive impairment are determinants of COVID-19 severity [36]. Another article by the same authors demonstrated that chronic neurological conditions, especially neurodegenerative diseases, increase lethality in COVID-19 cases [55]. In addition, Du et al. reported a 2.4-fold increase in mortality risk in patients with cerebrovascular or cardiovascular disease, although they did not specify the distinction between these two conditions [41]. 

Although in the present study, dementia was considered the second leading mortality cause among CNDs, it was not identified as a mortality marker. In this line, two systematic reviews and meta-analyses showed that the association between dementia and mortality in COVID-19 patients was affected by age and comorbidities [56,57]. Likewise, patients with Alzheimer’s disease and other types of dementia are at increased risk, given their advanced age, the coexistence of other comorbidities and poor cognition [58]. 

PD represented 6.5% of CND comorbidities and was equally distributed in phenotypes 1, 2 and 4 (primarily associated with other comorbidities, advanced age and respiratory symptoms). Parkinson’s disease-related mortality was 4.8%, and this CND was considered as a biomarker for best prediction of mortality. There is limited evidence on whether patients with PD are at increased risk of COVID-19 or worse outcomes. Our results are congruent with the first-mentioned systematic review and meta-analysis that concluded that the prevalence and prognosis of COVID-19 patients appear to be comparable in patients with PD and those without this condition. They postulated that the increased hospitalization and mortality might be attributed to advanced age and comorbidities [59].

In addition to clinical variables, laboratory examinations are also used as potential biomarkers to predict the degree of severity and mortality in patients with COVID- 19. It has been documented that hematologic biomarkers (lymphocyte count, neutrophil count and neutrophil–lymphocyte ratio (NLR)), inflammatory biomarkers (C-reactive protein (CRP), erythrocyte sedimentation rate (ESR)), immunological biomarkers (interleukin (IL)-6) and, especially, biochemicals (D-dimer, troponins, creatine kinase (CK)) as well as biomarker abnormalities can be used as predictive markers [47,60,61].

Furthermore, novel biomarkers such as homocysteine angiotensin II, Ang-(1–7), Ang-(1–9) and alamandine are under investigation [62,63]. On the other hand, other studies have shown a high frequency of autoantibodies such as antinuclear antibodies (ANAs), anti-antiphospholipid antibodies, and anti-cytoplasmic neutrophil antibodies (ANCAs) in patients with COVID-19 pneumonia, which seems to be associated with a poor prognosis [64,65,66]. This needs to be evaluated in larger case series in order to define their predictive clinical value as indicators of severe prognosis in COVID-19 patients.

## 5. Conclusions

To our knowledge, this is the first study thus far documenting the use of phenotypic clusters and the Stable Sparse Classifiers procedure to characterize the spectrum of COVID-19 cases and to predict mortality in ICU patients using more practical variables. The results from this study emphasize that a neurological symptom (ageusia), along with neurological diseases (cerebrovascular disease and PD) in addition to age, ICU days and specific symptoms (fever, cough, dyspnea and chilliness) and existing comorbidities (hypertension, diabetes and asthma) indicated the best prediction performance. The identification of clinical phenotypes and mortality biomarkers using robust statistical methodologies make several noteworthy contributions to basic and clinical investigations to distinguish the COVID-19 clinical spectrum and mortality. 

## 6. Limitations

The findings in this study are subject to at least two limitations. First, these findings are limited by the use of a retrospective cohort design. Secondly, the study did not include radiologic and laboratory examinations in the algorithm. Notwithstanding these limitations, the current study constitutes a novel approach to the heterogeneous clinical spectrum of COVID-19 using more feasible variables. 

Taking into account that a number of clinical studies are still ongoing both in Cuba and in many other countries, responses to treatment based on phenotypes could vary. Hence, this study might help update future randomized controlled trials of innovative therapies for COVID-19.

## Figures and Tables

**Figure 1 behavsci-12-00234-f001:**
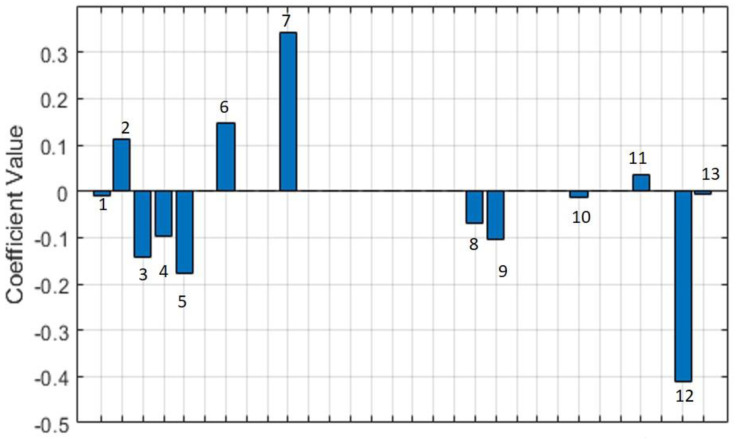
Selected variables which can be considered as biomarkers. According to the estimated model, the values obtained from the coefficient are presented on the y-axis. The index of each variable that was considered for the model design is shown on the x-axis. Bar 1—age, bar 2—sex, bar 3—fever, bar 4—cough, bar 5—dyspnea, bar 6—ageusia, bar 7—chilliness, bar 8—hypertension, bar 9—diabetes, bar 10—asthma, bar 11—cerebrovascular disease, bar 12—Parkinson’s disease, bar 13—ICU days.

**Figure 2 behavsci-12-00234-f002:**
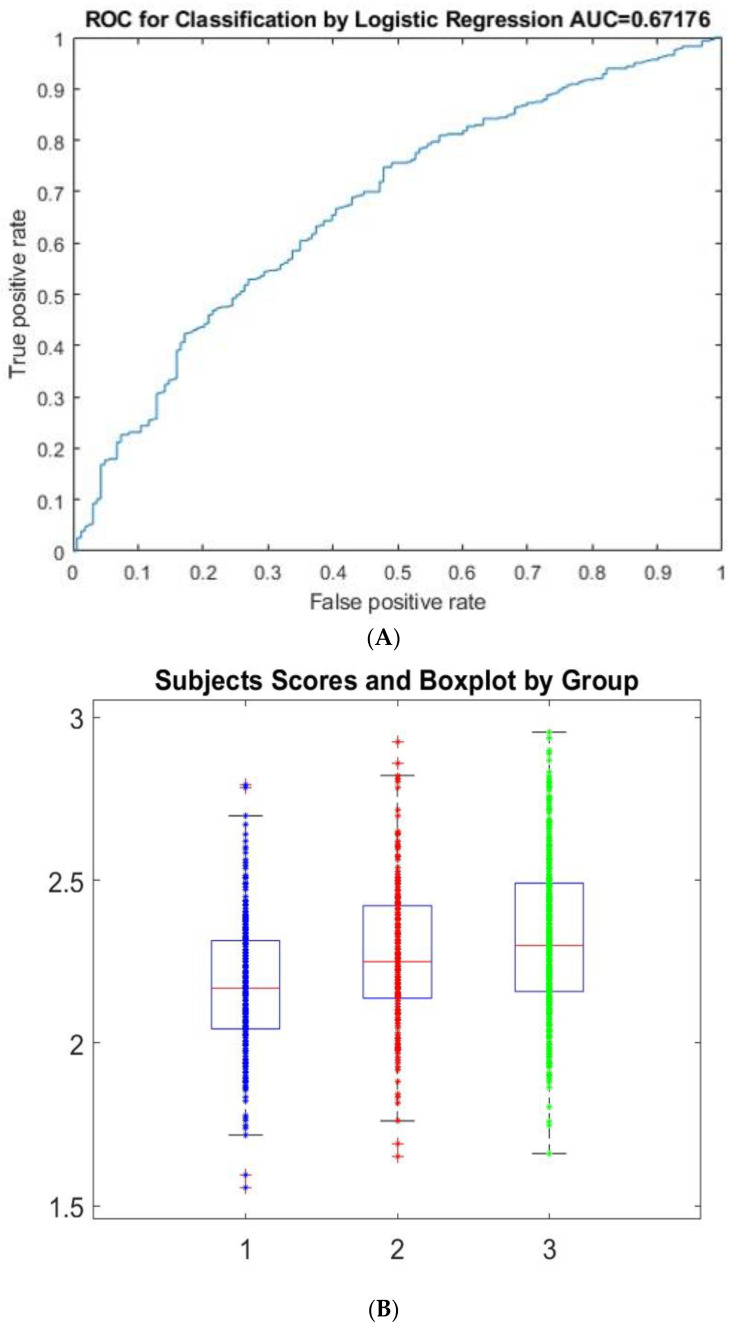
Performance of the predictor method applied to the 1252 COVID-19 patients in ICUs. (**A**) shows ROC curve for the global performance of the algorithm under the stability procedure. (**B**) is a boxplot of the individual classification according to the groups. As defined, the boxplot shows the mean, percentiles and dispersion of the groups. Note that Group 1 (deaths) is clearly distinguished from Groups 2 and 3 (no deaths); Group 2 (critically ill) and Group 3 (serious condition).

## Data Availability

The datasets produced for the current study are available from the corresponding author on reasonable request.

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
