# Peer review of "Clinical Phenotypes and Mortality Biomarkers: A Study Focused on COVID-19 Patients with Neurological Diseases in Intensive Care Units"

_behavsci, 2022, doi:10.3390/bs12070234_

Round 1
Reviewer 1 Report
In this study, the authors aimed at identifying clinical phenotypes and biomarkers for the best prediction of mortality considering age, symptoms and comorbidities focused on COVID-19 patients with neurological conditions in intensive care units. Data included 1252 COVID-19 patients admitted to ICUs to identify clinical patterns related to symptoms, comorbidities and age.
They identified six phenotypes: Phenotype class 1: mean age 72.3 years(ys)-- hypertension, coronary artery disease, alongside typical COVID-19 symptoms; class 2: mean age 63 27 ys - asthma, cough and fever; class 3: mean age 74.5 ys - hypertension, diabetes and cough; class 4: mean age 67.8 ys - hypertension and no symptoms; class 5: mean age 53 ys - cough and no 29 comorbidities; class 6: mean age 60 ys - without symptoms or comorbidities. Chronic neurological diseases (CND) percentage was distributed in the six phenotypes, predominantly in phenotypes class 3 (24.72 %) and 4 (35,39 %. χ² (5) 11,0129 p= ,051134. The cerebrovascular disease was concentrated in classes 3 and 4. χ² (5) = 36.63, p=, 000001. The mortality rate totaled 325 (25.79 %), of which 56 (17.23 %) had preexisting neurological diseases. The highest in-hospital mortality rates were found in phenotypes 1 (37. 22 %), and 3 (33.98%). The SSC revealed that a neurological symptom (ageusia) together with two neurological diseases (cerebrovascular disease and Parkinson’s disease) in addition to ICU days, age, specific symptoms (fever, cough, dyspnea, and chilliness) as well as particular comorbidities (hypertension, diabetes, asthma indicated the best prediction performance. They concluded that the identification of clinical phenotypes and mortality biomarkers make several noteworthy contributions to basic and experimental investigations for distinguishing the COVID-19 clinical spectrum and predicting mortality.
The study is of interest and of potential clinical impact. However, some points deserve further data and should be addressed.
-The study found a correlation between chronic neurological disease, clinical phenotypes, and in-hospital mortality in patients with COVID-19. Regarding the most frequent comorbidities, it is would be relevant also to specify if also atrial fibrillation was present in studied patients.
-How many patients (if any) were under steroid treatments?
-The authors should discuss clinically relevant literature data on other described prognostic markers of covid-19 patients. In particular, it has been reported that the positivity is antinuclear antibodies is significantly associated to severe lung disease in covid-19 patients has recently described (COVID-19 and Immunological Dysregulation: Can Autoantibodies be Useful? Clin Transl Sci. 2021 Mar;14(2):502-508; Antinuclear antibodies in COVID 19. Clin Transl Sci. 2021 Sep;14(5):1627-1628.).
Author Response
REVIEWER 1
questions |
Answer |
The study found a correlation between chronic neurological disease, clinical phenotypes, and in-hospital mortality in patients with COVID-19. Regarding the most frequent comorbidities, it is would be relevant also to specify if also atrial fibrillation was present in studied patients. - - |
No atrial fibrillation was precised in these patients.This sentence was included |
How many patients (if any) were under steroid treatments?
|
The following paragraph was included : Concerning treatment, 52,28 % of the patients received steroids, antibiotics (55,92%), anticoagulants (53,97%)and Jusvinza (Péptido CIGB (77,6%). AcMc (Itolizumab) and (Nimotuzumab) were also used in 150 of the cases. |
The authors should discuss clinically relevant literature data on other described prognostic markers of covid-19 patients. In particular, it has been reported that the positivity is antinuclear antibodies is significantly associated to severe lung disease in covid-19 patients has recently described (COVID-19 and Immunological Dysregulation: Can Autoantibodies be Useful? Clin Transl Sci. 2021 Mar;14(2):502-508; Antinuclear antibodies in COVID 19. Clin Transl Sci. 2021 Sep;14(5):1627-1628.).
|
The following paragraphs were included :
In addition to clinical variables, laboratory examinations are also used as potential biomarkers to predict the degree of severity and mortality in patients with COVID- 19. It has been documented that hematologic (lymphocyte count, neutrophil count, and neutrophil–lymphocyte ratio NLR) inflammatory (C-reactive protein (CRP), erythrocyte sedimentation rate (ESR), immunological (interleukin (IL)-6, and especially biochemicals (D-dimer,Troponins, creatine kinase (CK) as well as biomarker abnormalities can be used as predictive markers [47, 60-61].
Furthermore, novel biomarkers such as Homocysteine Angiotensin II, Ang-(1-7), Ang-(1-9), and alamandine are under investigation [62, 63]. On the other hand, other studies have shown a high frequency of autoantibodies such as antinuclear antibodies (ANAs), anti-antiphospholipid antibodies, and anti-cytoplasmic neutrophil antibodies (ANCAs) in patients with Covid-19 pneumonia, which seems to be associated with a poor prognosis [64-66]. This needs to be evaluated in larger case series in order to define their predictive clinical value as indicators of severe prognosis in COVID-19 patients.
These references were included
Giovanni Ponti, Monia Maccaferri, Cristel Ruini, Aldo Tomasi & Tomris Ozben (2020): Biomarkers associated with COVID-19 disease progression, Critical Reviews in Clinical Laboratory Sciences, DOI: 0.1080/10408363.2020.1770685 Núñez, Á.A. Priego-Ranero, H.B. García-González, et al.,Common hematological values predict unfavorable outcomes in hospitalized COVID-19 patients, Clinical Immunology (2021), https://doi.org/10.1016/j.clim.2021.108682 Hanff TC, Harhay MO, Brown TS, et al. Is there an association between COVID-19 mortality and the renin–angiotensin system—a call for epidemiologic investigations. Clin Infect Dis. 2020.
Liu Y, Yang Y, Zhang C, et al. Clinical and biochemical indexes from 2019-nCoV infected patients linked to viral loads and lung injury. Sci China Life Sci.2020;63(3):364–374. Chang SH, Minn D, Kim YK. Autoantibodies in moderate and critical cases of COVID19 [published online ahead of print Mar 10, 2021]. Clin Transl Sci. https://doi.org/10.1101/2021.03.09.434529. Pascolini S, Vannini A, Deleonardi G, et al. COVID-19 and immunological dysregulation: can autoantibodies be useful? Clin Transl Sci. 2021;14(2):502-508. Gao Z-W,Zhang H-Z,Liu C, Dong KE. Autoantibodiesin COVID-19:frequency and function. Autoimmun Rev.2021;20(3):102754.
|
Reviewer 2 Report
While I spent sometime reviewing this, the manuscript stands a great chance in excellence had it been the case with authros who could have spent more and illustrious time on background
The methods employed an dthe texts are just copy paste from various literature
Sorry for not being positive
I double checked whether the authors are same
Minor but essential
throughput is one word
preexisting may be two different words
L195: whom may be removed for 717
Pl correct to coccurences
pl make score as plural "scores" ( L385)
Author Response
REVIEWER 2
The article was revised exhaustively. Some aspects were reformulated. The Stable Sparse Biomarkers Detection procedure employed was described by Bosch-Bayard J, and Galán-García L,. The article was cited Bosch-Bayard J, Galán-García L, Fernandez T, Lirio RB, Bringas-Vega ML, Roca-Stappung M, Ricardo-Garcell J, Harmony T, Valdes-Sosa PA: Stable Sparse Classifiers Identify qEEG Signatures that Predict Learning Disabilities (NOS) Severity. Front Neurosci 2017, 11:749.(doi):10.3389/fnins.2017.00749. eCollection 02017.
Galan Garcia L is one of the author of the present study.
questions |
Answer |
Minor but essential throughput is one word
|
It was corrected |
preexisting may be two different words
|
Preexisting was replaced by chronic |
L195: whom may be removed for 717
|
It was corrected |
pl make score as plural "scores" ( L385)
|
It was corrected |
Reviewer 3 Report
Thank you for the opportunity to review this interesting manuscript, dealing with important findings entitled “Clinical phenotypes and mortality biomarkers: A study focused on COVID-19 patients with neurological diseases in intensive care units”. The article aims at identifying clinical phenotypes and biomarkers for the prediction of mortality considering age, symptoms, and comorbidities focused on COVID-19 patients with neurological conditions in intensive care units. Their statistical observations are interesting. The article includes a balanced and critical view of the findings in the field. These findings and explanations strategy is adequate and appropriate to minimize bias and errors. This article conforms to the relevant guidelines. The topic is covered objectively and based on sound empirical studies and explanations.
Author Response
Thank a lot
Best regards
Round 2
Reviewer 2 Report
The mansucript has been improved over all. The flow and breviy is much improved with major overhaul.
The statistical validtaion for six clinical phenotypes may subtly be mentioned
Figure 2 B in high resolution please
A few suggestions below:
L275: ARE reflected
L291: coefficientS
L357: In this age group